# Harnessing liquid-in-liquid printing and micropatterned substrates to fabricate 3-dimensional all-liquid fluidic devices

Wenqian Feng [1], Yu Chai[2,3], Joe Forth[1], Paul D. Ashby [1,3], Thomas P. Russell [1,4,5,6] & Brett A. Helms [1,3]

Systems comprised of immiscible liquids held in non-equilibrium shapes by the interfacial assembly and jamming of nanoparticle—polymer surfactants have significant potential to advance catalysis, chemical separations, energy storage and conversion. Spatially directing functionality within them and coupling processes in both phases remains a challenge. Here, we exploit nanoclay—polymer surfactant assemblies at an oil—water interface to produce a semi-permeable membrane between the liquids, and from them all-liquid fluidic devices with bespoke properties. Flow channels are fabricated using micropatterned 2D substrates and liquid-in-liquid 3D printing. The anionic walls of the device can be functionalized with cationic small molecules, enzymes, and colloidal nanocrystal catalysts. Multi-step chemical transformations can be conducted within the channels under flow, as can selective mass transport across the liquid—liquid interface for in-line separations. These all-liquid systems become automated using pumps, detectors, and control systems, revealing a latent ability for chemical logic and learning.

[1] Materials Sciences Division, Lawrence Berkeley National Laboratory, 1 Cyclotron Road, Berkeley, CA 94720, USA. [2] Department of Materials Science and Engineering, University of California, Berkeley, Berkeley, CA 94720, USA. [3] The Molecular Foundry, Lawrence Berkeley National Laboratory, 1 Cyclotron Road, Berkeley, CA 94720, USA. [4] Polymer Science and Engineering Department, University of Massachusetts, Conte Center for Polymer Research, 120 Governors Drive, Amherst, MA 01003, USA. [5] Beijing Advanced Innovation Center for Soft Matter Science and Engineering, Beijing University of Chemical Technology, 100029 Beijing, China. [6] WPI-Advanced Institute for Materials Research (WPI-AIMR), Tohoku University, 2-1-1 Katahira, Aoba, Sendai 980-8577, Japan. Correspondence and requests for materials should be addressed to B.A.H. (email: bahelms@lbl.gov)

The ability to shape immiscible liquids into prescribed architectures and reconfigure them on-demand is an emerging design paradigm in soft-matter materials chemistry[1–9]. To trap a liquid indefinitely in a nonequilibrium shape within another liquid, a suitable elastic film must be assembled at the liquid–liquid interface[4]. Assemblies of nanoparticle−polymer surfactants (NPSs), composed of colloidal nanoparticles originating from one of the liquids and polymer surfactants from the other, offer a compelling means both to assemble this elastic film and impart nanoparticle-derived functionality upon an all-liquid system[4]. NPS assembly at the interface leverages complementary chemical functionality at the surface of the nanoparticle and at the polymer surfactant chain end(s). This scheme is versatile, allowing the fabrication of reconfigurable systems of discrete liquid droplets[10,11], long liquid threads[6,12], molded objects[13], and bicontinuous interfacially jammed emulsions[5]. Still nascent in their development, structured liquids presently lack clear design rules for controlling their 2D or 3D architectures, spatially directing functional components within each liquid phase, and coupling physiochemical processes across the liquid−liquid interface so as to create autonomous chemical systems capable of performing useful work, processing information, or executing logical functions.

Here, we harness the structured liquid paradigm to fabricate 3D all-liquid fluidic devices that are infinitely reconfigurable and endowed with spatially programmable functions. In turn, we gain insights into their potential for rendering chemical systems of arbitrary complexity to perform tasks, including chemical separations, multi-step chemical transformations, and chemical logic.

## Results

**All-liquid fluidic device fabrication**. To fabricate all-liquid fluidic devices, glass supports are first coated with a super-hydrophobic polymer[7] before undergoing photo-patterning with a variety of superhydrophilic channel architectures (Fig. 1a). A polar phase—e.g., an aqueous dispersion of anionic 2D nanoclays (10 mg mL$^{-1}$ at pH 7.0)—is then deposited onto the super-hydrophilic region prior to immersion in an oil phase—e.g., a solution of homotelechelic amine-terminated poly(dimethylsiloxane) (H$_2$N-PDMS-NH$_2$, $M_n$ ~27 kg mol$^{-1}$, 10% w/w) in either toluene, silicone oil, or dodecane (Supplementary Fig. 1). Interfacial forces are effective in pinning and confining the aqueous phase in arbitrarily complex geometries and a wide range of channel widths (635–3000 μm) (Supplementary Fig. 2).

Once the liquid phases are in place on the patterned glass support, NPS film formation at the water−oil interface is extremely rapid[1,14,15]. Its kinetic trajectory was monitored via time-evolution of interfacial tension (IFT) for pendant droplets of 2D nanoclay dispersions (0.5–10 mg mL$^{-1}$) within an oil phase consisting of H$_2$N-PDMS-NH$_2$ (10% w/w) in toluene (Fig. 1b). The concentration of nanoclay influences the rate of diffusion to the interface, the ionic strength of aqueous phase and the areal density at the interface at steady-state (Supplementary Fig. 3a); the latter two ultimately dictate the IFT[16,17]. In all cases, the system reached steady-state rapidly. Furthermore, NPS assembly into an elastic film upon interface compression was irreversible, as evidenced by the film's wrinkling behavior upon droplet retraction (Fig. 1c, Supplementary Fig. 3b and 4, and Supplementary Movie 1). The topography of the NPS film at the water–oil interface was revealed using in-situ atomic force microscopy (AFM)[16] (Fig. 1d). Here, we immersed the AFM tip in the aqueous phase and brought it into contact with the water–oil interface, where a tiny drop of silicon oil (viscosity $\eta = 60{,}000$ cSt) with 10% w/w of H$_2$N–PDMS–NH$_2$ had been pinned on the silicon substrate and subsequently enshrouded by a dispersion of 2D nanoclays (10 mg mL$^{-1}$). The data showed well-packed nanoclays at the liquid–liquid interface: no structural aberrations were visible.

NPS films exhibit the necessary mechanical strength and elasticity to render patterned all-liquid microchannels shape-persistent under high flow rates, even at sharp curves defined by superhydrophilic surface patterns: e.g., in U-bends of serpentine channels (Fig. 1). 2D nanoclay particles are held strongly to the interface by the surfactant. Incomplete packing of nanoclays yields pores that expose surfactant capable of binding a second nanoclay layer on top of the first (Fig. 1d). These nanoclay–nanoclay interactions ultimately determine the mechanical properties of the NPS films for small displacements both compressive and extensive. To quantify the elasticity of NPS films, we compressed them uniaxially in a planar geometry in a Langmuir−Blodgett trough (Supplementary Fig. 5). The wavelength of the wrinkles, $\lambda$, is directly related to the bending modulus, $B$, by $\lambda = 2\pi \left(\frac{B}{\Delta\rho g}\right)^{\frac{1}{4}}$[18,19], where the density difference between oil and water is $\Delta\rho = 70$ kg m$^{-3}$, and $g = 9.8$ m s$^{-2}$. $\lambda$ was obtained by image analysis of 2748 wrinkles to be $\lambda = 113 \pm 22$ μm, giving a bending modulus on the order of $10^4$ $k_\mathrm{B}T$ for the NPS film—at least two orders of magnitude larger than that typically measured in nanoparticle assemblies at a liquid–liquid interface[20,21]. To demonstrate the value these elastic properties bring to all-liquid fluidic devices, we pumped an aqueous solution of resazurin dye (50 μg mL$^{-1}$, pH 7.0) through a channel ~139 mm in length and ~2 mm in width at a flow rate of 10 mL h$^{-1}$, significantly faster than the maximum reported in liquid microfluidics configurations that lack a functional interfacial membrane (Fig. 1e, f). Only patterned microchannels bearing NPS walls were capable of guiding flow along the channels at these rates without aberrations to the all-liquid architectures; absent these elastic NPS walls, the aqueous phase irreconcilably accumulates a volume at the entry point of the channel (Supplementary Movie 2). Thus, while all-liquid fluidics have, in the past, made use of surface patterning and surface adhesion to produce complex all-liquid structures[22–25], these schemes relied on surface tension and contact angle hysteresis[26,27] for pumping one fluid within the other, passively or actively. This set limits for the types of fluids and the flow rates that can be used, and restricted the chemistries that can be loaded into each phase. In our scheme, rather than flow being driven by Laplace pressure, contact angle hysteresis or electric field strength[28], the internal overpressure under flow is balanced by the elastic NPS wall. As a result, all-liquid fluidic systems may now be used at high flow rates (up to 10 mL h$^{-1}$) at markedly reduced surface tensions and, notably, in the presence of surface-active species. The maximum flow rate of all-liquid fluidic devices will of course depend both on the cross-sectional area of the channel as well as the channel architecture, and can be explored endlessly as such.

**Membrane permeability**. The irregular shapes of nanoclays yield NPS films with residual interstitial spaces, if the assemblies are confined to 2D, which are selectively permeable to solutes in either liquid phase[29]. Overlapping nanoclays may also contribute to the film's observed permeability. We demonstrated selective mass transfer across the interface (Fig. 2a) by carrying out crossover experiments using a charge-neutral fluorescent dye (dye **1**, Fig. 2b and Supplementary Fig. 6) that has an affinity for both water and toluene, and compared the crossover rate with those of anionic resazurin (**2**) and cationic rhodamine 6G (**3**). We introduced a solution of H$_2$N-PDMS-NH$_2$ in silicone oil

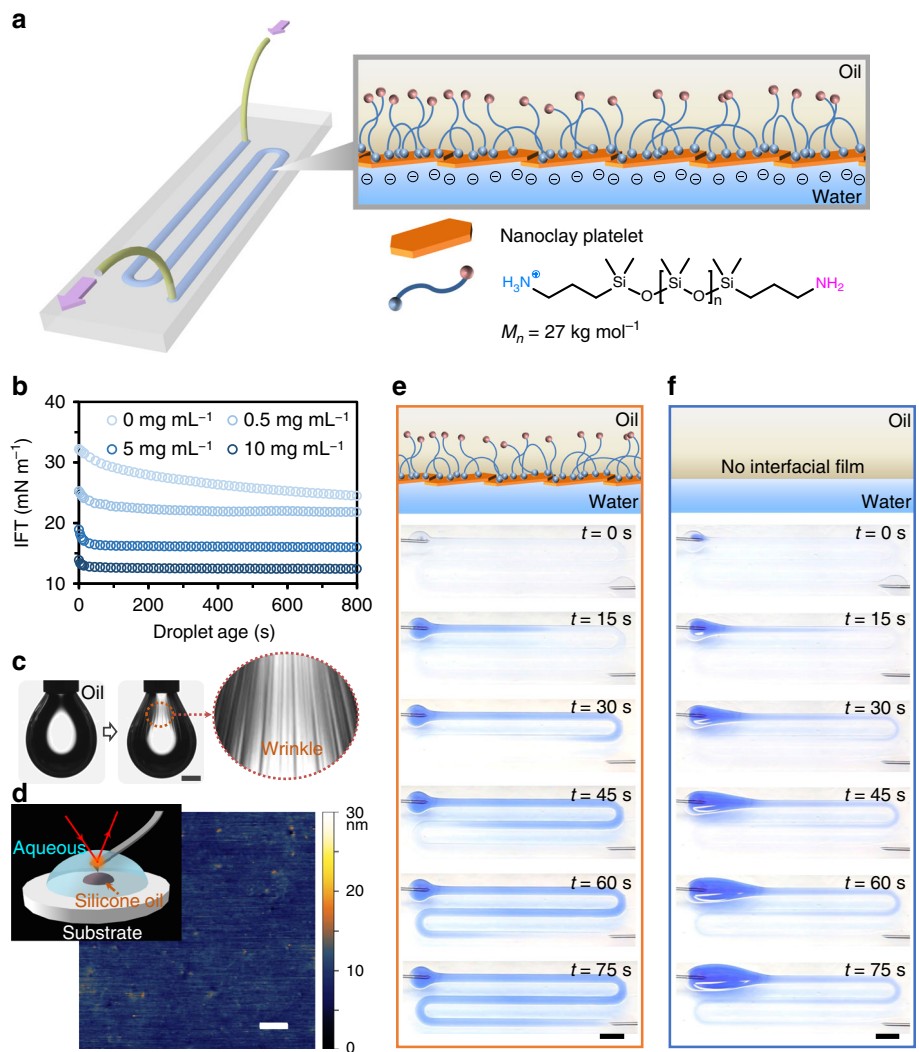

**Fig. 1** All-liquid fluidic devices stabilized by NPS membranes. **a** Schematic of an all-liquid fluidic device comprised of immiscible liquid phases confined in space using superhydrophobic—superhydrophilic micropatterned substrates. Nanoclay–polymer surfactants (NPSs) self-assemble at the liquid–liquid interface, forming an elastic wall that allows the all-liquid architecture to maintain integrity while fluid is pumped through the channel. **b** Temporal evolution of interfacial tension (IFT) of aqueous nanoclay dispersions (0.5, 5, and 10 mg mL$^{-1}$, pH 7.0) introduced to solutions of H$_2$N-PDMS-NH$_2$ in toluene (10% w/w), illustrating control over the rate of NPS assembly at the interface. **c** Buckling behavior observed when retracting a droplet cladded with the interfacial NPS film. Scale bar, 1 mm. **d** In-situ AFM image of the NPS film. The inset shows the schematic diagram of the experimental setup for AFM measurements. Scale bar, 100 nm. **e**, **f** Time-lapse images of a solution of blue-colored dye being pumped (10 mL h$^{-1}$) through the channel in the presence (**e**) and in the absence (**f**) of the NPS film. Scale bars, 5 mm. AFM atomic force microscopy

(10% w/w) into a quartz cuvette containing a dispersion of nanoclays (9 mg mL$^{-1}$) and dye (100 μg mL$^{-1}$) and let the system form an NPS interfacial film; owing to the insolubility of dyes in silicone oil, these reporters remain barricaded inside the aqueous phase. We then added toluene to the oil phase and observed the rate at which dye **1** partitioned into the toluene phase by monitoring the optical absorbance of the aqueous phase (Fig. 2b, Supplementary Fig. 7 and 8); the decrease in optical absorption indicated successful partitioning of the dye into the organic phase. In contrast, neither of the ionic dyes partitioned into the organic phase (Supplementary Fig. 7). The partitioning of non-ionic dye **1** was further confirmed in the patterned all-liquid device using laser scanning confocal microscopy (LSCM) (Fig. 2c and Supplementary Fig. 9); our success in this format suggested that the semi-permeable NPS film could be utilized for in-line chemical separations under flow. To demonstrate this feature, a solution of dyes **1** and **2** was pumped through the channel (0.2 mL h$^{-1}$), which resulted in a color change after ~1 h under continuous

flow, both within the aqueous channel as well as in the overlaid toluene (Fig. 2d and Supplementary Movie 3). More than 93% of dye **1** was separated from the aqueous dye mixture by liquid–liquid partitioning into the toluene phase; dye **2** was retained in channel (Fig. 2e and Supplementary Fig. 10).

**Membrane functionalization.** Beyond their use as structural elements, anionic nanoclays further allow cationic molecules, enzymes, and nanoparticles to be anchored to the NPS wall (Fig. 3a–c). In turn, all-liquid fluidic devices are primed for biphasic chemical synthesis. We were initially drawn to this possibility by noting changes in the optical absorption of rhodamine 6G in the presence of nanoclays during the crossover experiments: the molar extinction spectrum of cationic dyes changed over time, consistent with adsorption to the anionic nanoclays (Supplementary Fig. 7), whereas no such changes were observed for anionic or non-ionic dyes. We confirmed that

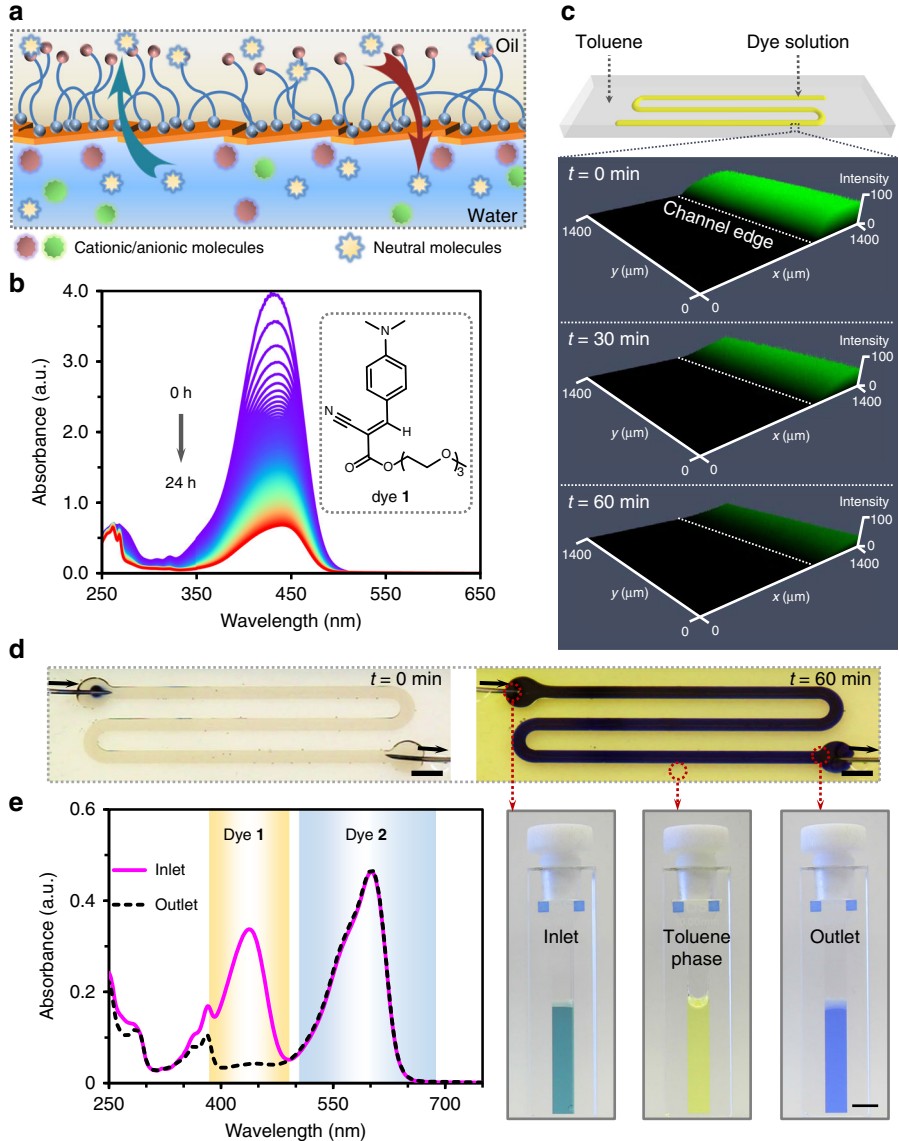

**Fig. 2** Semi-permeability of 2D NPS interfacial films. **a** Schematic showing selective mass transfer across the water–oil interface through the NPS film. **b** Stacked UV–Vis spectra of a neutral dye (**1**) at 5-min time-intervals, monitoring dye transfer from water to toluene across the NPS membrane film in a masked cuvette. The inset shows the chemical structure of dye **1** engaged in mass transfer. **c** LSCM images of the fluidic channel after infusing dye **1** solution in the preformed liquid channel covered by toluene. **d** A fluidic device depicting the flow of a mixed solution of dye **1** and resazurin sodium (**2**) for chemical separation. [**1**] = 1.4 mg mL$^{-1}$, [**2**] = 1 mg mL$^{-1}$. Photographs of the diluted inlet, outlet, and overlay of toluene solutions collected in cuvettes show the dye separation after infusing the mixed solution at a flow rate of 0.2 mL h$^{-1}$ for 1 h. Scale bar, 5 mm. **e** UV–Vis spectra of the diluted inlet and outlet solutions used to quantify the efficiency of dye **1** partitioning from the aqueous phase to the toluene phase. NPS nanoparticle—polymer surfactant, LSCM laser scanning confocal microscopy

cationic small molecules were immobilized on the NPS walls by using LSCM to visualize the spatial distribution of methylene blue (Fig. 3d and Supplementary Fig. 11) and rhodamine 6G (Supplementary Fig. 12) in the device. After pumping cationic dyes into the channel, aging the system such that they diffuse to the interface, and rinsing with water to remove unbound objects, the immobilized fluorophores each tracked the curvature of the NPS wall. We were similarly able to anchor to the wall cationic enzymes (e.g., horseradish peroxidase, HRP) as well as cationic nanocrystal catalysts (e.g., cetyltrimethylammonium bromide (CTAB)-modified Pt nanocrystals, $\zeta = +19.2 \pm 1.3$ mV). While the fluorescence of adsorbed dye-labeled HRP (lissamine rhodamine B sulfonyl chloride (LRSC)-labeled HRP) was readily visualized using LSCM (Fig. 3e and Supplementary Fig. 13), the

immobilization of CTAB-Pt nanocrystals with respect to the underlying nanoclay tiles at the liquid–liquid interface required in-situ AFM to properly visualize nanocrystal density (Fig. 3f, and Supplementary Fig. 14 and 15). Notably, we did not observe any nanocrystal mobility at the NPS surface in the in-situ AFM experiments, even upon repeated imaging over a fixed area; the nanocrystals are well-anchored to the nanoclays.

**All-liquid microreactor.** We carried out flow-driven multistep chemical transformations using all-liquid fluidic devices configured with HRP@NPS walls (Supplementary Fig. 16 and 17). In a first demonstration, we pumped a solution of 3,3′,5,5′-tetramethylbenzidine (TMB) and hydrogen peroxide (Fig. 3g) through

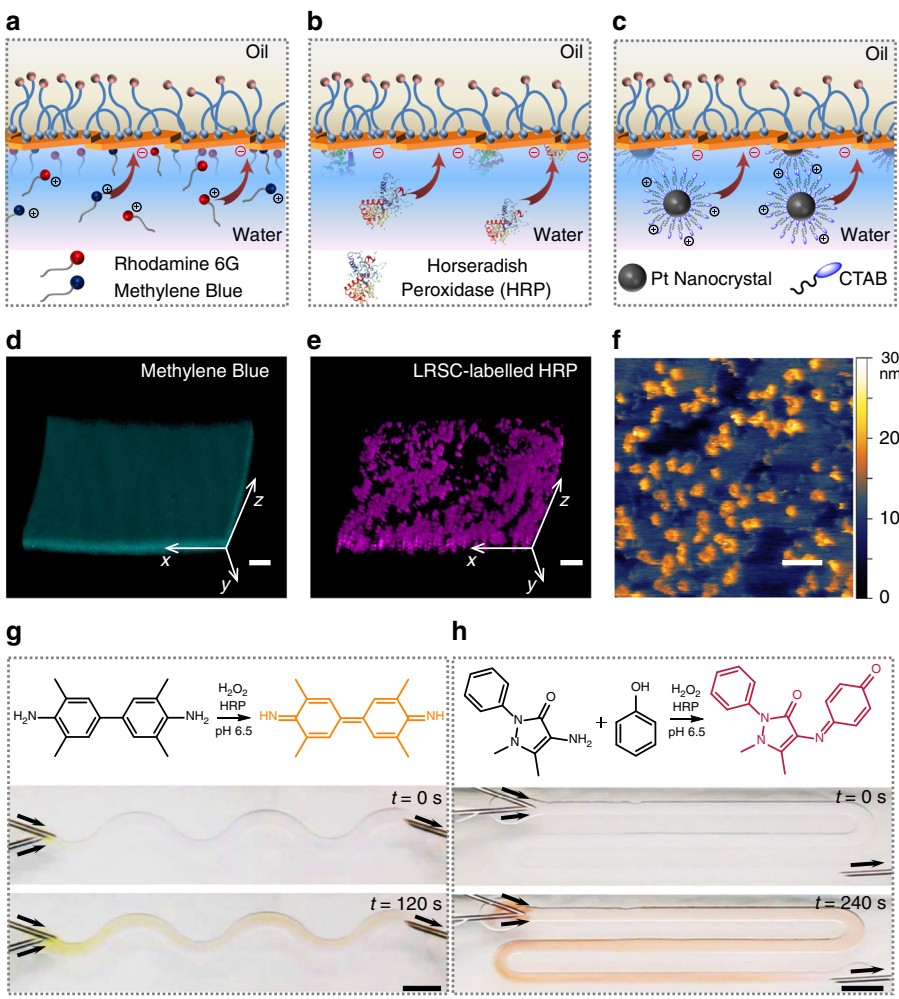

**Fig. 3** All-liquid microreactors. **a–c** Schematic of the anchoring of cationic molecules (**a**), enzymes (**b**), and nanocrystals (**c**) to the anionic NPS film lining the microchannels. **d, e** 3D reconstitution of confocal images of methylene blue (**d**) and LRSC-labeled HRP (**e**) to determine the adhesion of cationic species to the NPS wall of the microchannel. Scale bars, 100 μm. **f** In-situ AFM image of the NPS film immobilized with CTAB-coated Pt nanocrystals at the water−silicone oil interface. Scale bar, 100 nm. **g, h** Oxidation of TMB (**g**) and 4-AAP/phenol (**h**) substrates catalyzed by immobilized HRP in the channels, which generate products with distinctive colors under flow. Scale bars, 5 mm. NPS nanoparticle−polymer surfactant, LRSC lissamine rhodamine B sulfonyl chloride, HRP Horseradish peroxidase, TMB 3,3′,5,5′-tetramethylbenzidine, 4-AAP 4-aminoantipyrine

the channel (1 mL h⁻¹), which led to the oxidation of TMB to yield a yellow-colored diimine product[30] (Supplementary Fig. 18). We visually tracked the trajectory of this reaction along the channel length, as the substrate, intermediate, and product are chromogenically distinct. In a second demonstration, we pumped a solution of phenol, 4-aminoantipyrine (4-AAP) and hydrogen peroxide (Fig. 3h) through the channel (1 mL h⁻¹), which led to the oxidative coupling of phenol and 4-AAP to yield a red-colored quinone-imine conjugate[31]. While this second sequence is more complex, the kinetics were fast, likewise allowing the reaction's trajectory to be visualized in the channel via the intense color of the final product. Given our successes with these reactions, we next implemented an expanded set of chromogenic oxidative coupling reactions to demonstrate how the spatial programmability of function should be coupled to active programming and re-programming of the flow paths of the device to create a type of chemical logic.

**Programmable, reconfigurable, and self-healing microreactors.**
An advantage to using NPS films to stabilize all-liquid fluidic devices is that additional microchannels are straightforward to

introduce via direct-write methods (e.g., 3D liquid-in-liquid printing), even after the device has been configured and its local chemistry defined by the user (Fig. 4). To contextualize this advance, without the aid of micropatterned substrates and NPS films, constructing channel-like aqueous threads in oil is not typically possible, as the thermodynamic driving force required to reduce interfacial area breaks up aqueous threads into droplets[32]. In that wall-forming is fast, and the jammed assembly counteracts in a timely manner the compressive force in the system arising from Plateau–Raleigh instabilities:[6,12,33] both patterned 2D channels and 3D printed inter-channel microtubes were stable for at least 2 h, i.e., while devices were in use. The stability of our printed microchannels allows them to be used as bridges between different regions on the substrate, or to couple the device to an exogenous entity (Fig. 4a–d).

We found that despite the perturbative nature of anchoring, printing, and bridging liquid microchannels, NPS films self-heal on a time-scale of ms-to-s (Supplementary Fig. 19). This allows spatially programmed functionality to be connected in a user-defined sequence in real time. Printed channels retained their structure during flow, as demonstrated by the pumping of aqueous blue dye through three liquid regions connected by two

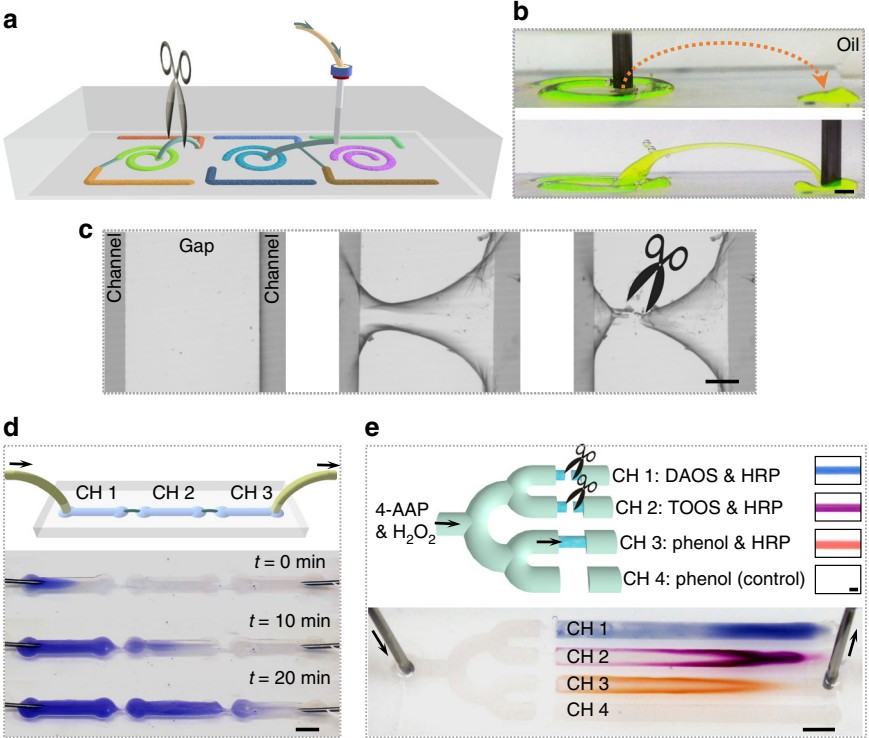

**Fig. 4** Reconfigurable 3D all-liquid microreactors with logical optical outputs. **a** Schematic of fabricating 3D microchannels stabilized by nanoclay-surfactant interfacial assembly to connect the separated microchannels by printing of water in oil. Disconnecting these microchannels is also possible by cutting these "bridges" off. **b** Adjacent channels on a patterned substrate connected by a 3D printed overpass. Na-fluorescein (0.1 mg mL$^{-1}$) has been added to the clay dispersion to clarify features of the constructs. Scale bar: 2 mm. **c** Fabrication of a 3D bridge linking separate channels (middle panel) and clipping the bridge (right panel). Scale bar: 500 μm. **d** Time-lapse showing a dyed solution pumped through liquid channels CH 1, CH 2, and CH 3, which are connected by printed 3D bridges. Scale bar: 5 mm. **e** Chemical transformations yielding chromogenically distinct products are conducted consecutively by forming and then breaking bridges between the inlet and various patterned channels containing different reagents. The bridges are built using liquid-in-liquid 3D printing to yield a "turn-on" state, and mechanically broken to yield a "turn-off". Inset photographs on top show the solutions collected in capillaries after carrying out these chromogenic reactions in vials. DAOS: N-ethyl-N-(2-hydroxy-3-sulfopropyl)-3,5-dimethoxyaniline, sodium salt. TOOS: N-ethyl-N-(2-hydroxy-3-sulfopropyl)-3-methylaniline, sodium salt. Scale bars: 1 mm (top) and 5 mm (bottom)

liquid bridging arcs (Fig. 4d). We were also able to clip these 3D bridges (Fig. 4c), allowing once-connected patterned regions to be uncoupled on-demand and to re-direct the flow to other channels patterned on the chip. As a proof-of-concept integration of these concepts into spatially programmable chemical logic, we used a fourfold divergent set of channels that each required a unique printed liquid thread for permitting flow through them. Within a given path (i.e., channels 1–3), we integrated HRP and one of three different oxidative coupling partners—DAOS, TOOS, and phenol—for the 4-AAP substrate; in the fourth channel, we did not include the coupling partner (negative control). We then connected each of these channels sequentially to the device's inlet of 4-AAP and $H_2O_2$, and configured an outlet for the effluent while pumping; between each spatially guided reaction in a desired channel, the previous liquid bridge was clipped so that flow was only directed in the selected channel. We then visualized the chemistries in all channels as visual indicators for how the flow was directed and re-directed to different channels by the user over the course of the experiment via the distinctive hues of the various oxidatively coupled products in those regions after exposure to the complementary substrates (Fig. 4e, Supplementary Figs. 20 and 21).

## Discussion

The ability to spatially program and re-program on-demand the functionality and geometric layout of a fluidic device, including channel compartmentalization, flow paths between channels, and extent of chemical coupling between the immiscible fluids comprising the device opens exciting directions in responsive soft-matter chemical systems. Our studies uncover a latent learning ability in such devices, in that physiochemical sensing or detection of channel properties and contents can be used to direct the architecture of the device to achieve a specific outcome. Maturation of the design concept led to devices that can execute complex tasks in a logical manner by reversible compartmentalization of function and direction of chemo-energetic flows that operate far from equilibrium conditions[34–37]. The potential for this system to exhibit autonomous learning is evident. Such devices may also be arrayed to generate deep or dark data[38] for machine learning, e.g., from all-liquid (bio)chemical transformations and screens, to build knowledge and understanding from chemical logic.

## Methods

**Fabrication of photopatterned glass substrates**. We employed a previously published protocol[7] with slight modifications (Supplementary Figure 1). Briefly, 25 μm-thin strips of Teflon film (American Durafilm Co.) were placed at the edges of one 3-(trimethoxysilyl)propyl methacrylate-modified glass plate (76.2 × 25.4 × 1 mm, VWR Microscope Slides) and one fluorinated glass slide was clamped on top of it. A polymerization mixture consisting of 2-hydroxyethyl methacrylate (24% w/w), ethylene dimethacrylate (16% w/w), 1-decanol (12% w/w), cyclohexanol (48% w/w) and 2,2-dimethoxy-2-phenylacetophenone (photoinitiator, 1% w/w with respect to monomers) was injected into the mold between the glass slides and irradiated for 15 min with 12.0 mW cm$^{-2}$ UV light (260 nm). After

removing the glass cover, the newly formed polymer layer was washed extensively with ethanol. The superficial surface was removed by using an adhesive film ("Scotch tape"). The polymer layer was then immersed into dichloromethane (50 mL) containing 4-pentynoic acid (112 mg, 1.14 mmol) and 4-(dimethylamino)pyridine (DMAP, 56 mg, 0.46 mmol) and N,N′-diisopropylcarbodiimide (176.5 μL, 1.14 mmol), followed by stirring the solution at room temperature for 4 h. The plate was washed extensively with acetone and dried using a stream of nitrogen.

Surface patterning was performed using thiol-yne reactions. The polymer layer was wet with ethyl acetate solution containing 10 vol% 1 H,1 H,2 H,2H-perfluorodecanethiol, covered by a photomask, and irradiated by UV light (260 nm) for 30 s. After removing the photomask, washing with acetone and drying, the polymer layer was wet with an ethanol−water mixture (1:1 v/v) containing 2-mercaptoethanol (15% v/v) and 2,2-dimethoxy-2-phenylacetophenone (1% w/w). Finally, the polymer layer was irradiated by UV light (365 nm, 2 mW cm⁻²) for 120 s, and washed extensively with ethanol followed by drying. Static water contact angles (WCAs) were measured after each modification (Supplementary Fig. 1b).

Modification of the surface with 1 H,1 H,2 H,2H-perfluorodecanethiol endows the polymer surface with superhydrophobicity (static WCA being as high as 165.1°), as functionalization of the surface with 2-mercaptoethanol transforms the polymer into a superhydrophilic surface (static WCA = 4.8°). Due to the extreme difference in wettability of the superhydrophilic area compared to the superhydrophobic barriers, water is spontaneously repelled by the barriers, but fills the superhydrophilic regions.

**All-liquid device fabrication**. An appropriate volume of the nanoclay dispersion (10 mg mL⁻¹) was applied to each superhydrophilic area to in-fill the pattern's contour. The entire surface was then covered with a solution of $H_2N$-PDMS-$NH_2$ surfactant in toluene, silicone oil, or hexadecane (10% w/w), leading to the formation of nanoclay–polymer surfactant (NPS) assembly at the liquid–liquid interface. Stainless steel needles (22-gauge, 0.413 mm inner diameter) affixed to syringes, which were attached to syringe pumps, were inserted into the inlet and outlet arms of the circuits (Supplementary Fig. 1c) to complete the device. Plastic or other needles are also applicable here, and needles with hydrophilic surfaces are preferred to realize the most effective seals.

Using different shadow masks, we generated superhydrophobic–superhydrophilic micropatterns with custom-purposed geometries, and utilized them for tailoring the functions of our all-liquid fluidic devices. Aqueous solutions are pumped through microchannels with different widths, and withdrawn by external pumps simultaneously, to achieve flow through the entire structure (Supplementary Fig. 2). We find optimum pH ranges to be 5.0–9.0 for aqueous fluids pumped at optimum flow rates of 0.15–10 mL h⁻¹ for channel dimensions of 300–2000 μm.

**Liquid–liquid interfacial tension measurements**. Interfacial tension measurements were conducted using pendant drop tensiometry. Droplet profiles were analyzed using Krüss DSA25 Drop Shape Analyzer, with IFTs calculated using the DSA Advance software after fitting the data to the Young–Laplace equation.

**Elasticity of NPS films**. Compression experiments were performed in a purpose-built Delrin Langmuir–Blodgett trough to test the bending modulus of the NPS interfacial assembly. The trough had a lip at which the water–oil interface was pinned, allowing for observation of the effects of NPS compression on an approximately planar liquid–liquid interface. The aqueous phase (pH 7.0, containing 10 mg mL⁻¹ laponite nanoclays) was added; silicone oil (viscosity $\eta = 10$ cSt) containing 10% w/w PDMS-$NH_2$ ($M_w = 27$ kg mol⁻¹) was then carefully pipetted into the trough side-wall above the aqueous phase, minimizing disturbances to the water–oil interface. After ageing the system for 15 min, uniaxial compressions were performed manually using a linear translation stage. To obtain wrinkle wavelengths from the images, intensity profiles of a line thickness of 20 pixels were convolved with a 3-pixel-width Gaussian kernel to remove high-frequency noise using the scikit-image Python module, and peaks detected using the peakutils Python module. Final values for wavelength are the average of 2748 peak separations (Supplementary Fig. 5).

Laser scanning confocal fluorescence microscopy (LSCM) revealed sinusoidal wrinkles with wavelength, $\lambda \approx 150$ μm in the NPS assembly at the water–oil interface (Supplementary Fig. 12c). These wrinkles are characteristic of dilatational stresses acting on the NPS film, and their wavelength is directly related to the elastic moduli of the film and the geometry of the system[18,19,39]. Quantitative information was extracted from these wrinkles by performing uniaxial compression experiments on them in a Langmuir–Blodgett trough. Upon compressing the film, sinusoidal wrinkles of $\lambda = 113 \pm 22$ μm were observed, in reasonable agreement with the value observed using LSCM. The slight difference was attributed to the different measurement geometry, though the different stress histories in the NPS assemblies also play a role. Applying the relationship $\lambda = 2\pi \left(\frac{B}{\Delta \rho g}\right)^{\frac{1}{4}}$, where the density difference between the oil and the water, $\Delta \rho = 70$ kg m⁻³, and $g = 9.8$ m s⁻² gives a bending modulus $B = 1.8 \times 10^4 \, k_B T$. Using the relationship $B = \frac{Ed^3}{12(1-\nu^2)}$, where $d \approx 10$ nm is the film thickness (measured from the AFM image in Fig. 3f) and the

Poisson ratio, $\nu \approx \frac{1}{3}$, the Young's modulus, $Y_{3D} \approx 0.5$ GPa, comparable to values obtained by direct measurements on nanoparticle assemblies[20,21]. The 2D Young's modulus, $Y_{2D} \equiv Y_{3d}d \approx 5$ N m⁻¹ is fully three orders of magnitude larger than the water–oil IFT as measured in Fig. 1b. These results are remarkable for two reasons. First, our ultra-thin NPS assembly has bending moduli orders of magnitude larger than those measured previously in the literature[19,40] or predicted by theory[21]. We attribute this to both the overlapping structure of the nanoclay assembly and the steric interactions between the bound polymer brushes that reinforce the film. Second, the large Young's modulus of the film underscores the enhanced structural robustness and higher flow rates that the film allows us to achieve in comparison to methods in which pumping and structure are derived solely from surface forces.

**Synthesis of ambipolar dye 1**. The synthesis of dye **1** (Supplementary Fig. 6a) was carried out over two steps. Briefly, to a dichloromethane solution (20 mL) of cyanoacetic acid (10 mmol), triethylene glycol monomethyl ether (10 mmol), and DMAP (1 mmol), a solution of dicyclohexylcarbodiimide (10 mmol) in dichloromethane (10 mL) was added dropwise at 0 °C. The mixture was allowed to warm to room temperature and was stirred for 2 h. After filtering the solid byproduct in the mixture under gravity, the filtrate was dried using $MgSO_4$, concentrated and purified by column chromatography (silica, 10–80% ethyl acetate in hexane) to obtain a pure 2-(2-(2-methoxyethoxy)ethoxy)ethyl 2-cyanoacetate (71% yield). Next, to a solution of 4-(dimethylamino)benzaldehyde (3.35 mmol) and 2-(2-(2-methoxyethoxy)ethoxy)ethyl 2-cyanoacetate (3.35 mmol) in THF (7.5 mL) was added 1,8-diazabicyclo[5.4.0]undec-7-ene (DBU) (3.40 mmol) in THF (0.5 mL). The solution was concentrated under vacuum after stirring for 1 h at room temperature. Dye **1** was purified via silica gel chromatography (10–100% ethyl acetate in hexanes) (68% yield). ¹H NMR (500 MHz, CDCl₃): δ 8.08 (s, 1H), 7.95 (d, 2H, J = 9.0 Hz), 6.71 (d, 2H, J = 9.1 Hz), 4.43 (m, 2H), 3.83–3.81 (m, 2H), 3.73–3.75(m, 2H), 3.70–3.67 (m, 4H), 3.58–3.56 (m, 2H), 3.39 (s, 3H), 3.12 (s, 6H) ppm. These data were consistent with a previous report[41]. Dye **1** dissolves in water and toluene, but not in silicone oil. This molecular rotor shows differentiated fluorescence in water and in toluene (Supplementary Fig. 6b).

**Permeability of NPS films at the liquid–liquid interface**. Mass transfer across the NPS interfacial assembly in a cuvette (Supplementary Fig. 7a) was monitored using 5 μL of dye **1**, resazurin (dye **2**), or rhodamine 6G (dye **3**) (each 1 mg mL⁻¹ in water). The dyes were combined with 45 μL nanoclay dispersion (10 mg mL⁻¹, pH 7) in the cuvette, and then covered with 50 μL solution of $H_2N$-PDMS-$NH_2$ (10% w/w in silicone oil (viscosity $\eta = 10$ cSt, density $\rho = 930$ kg m⁻³ at 25 °C). After 15 min of aging to form the NPS film, UV–Vis spectra were recorded between 250 and 650 nm at 5-min time-intervals after gently adding 1.5 mL toluene solution of $NH_2$-PDMS-$NH_2$ (10% w/w) in the cuvette (Supplementary Fig. 7). Adsorption of dye **1** to the nanoclay is negligible (Supplementary Fig. 8).

To confirm the semi-permeable character of the NPS wall in all-liquid fluidic devices, dye **1** (1 mg mL⁻¹) in a dispersion of nanoclays (10 mg mL⁻¹) was introduced to a microchannel bearing an NPS interfacial film. The channel was interrogated by LSCM and the field of view set at the edge of the channel. Fluorescence (excitation, 405 nm; detection range, 510–600 nm) from the dye shown in green indicated the channel position (Supplementary Fig. 9a). Loss of fluorescent intensity in the channel area was observed over time, illustrating dye transfer from water to toluene (Fig. 2c, Supplementary Fig. 9b, c). Due to the weaker emission of the dye in toluene at this wavelength relative to that in water, once the dye migrates into the toluene, it is effectively masked, allowing very careful measurements of the rate of outward diffusion.

**Chemical purification by liquid–liquid partitioning**. An aqueous nanoclay dispersion (10 mg mL⁻¹, pH 7) containing ambipolar dye **1** (1.4 mg mL⁻¹) and anionic dye **2** (1 mg mL⁻¹) was pumped through the channel (~139 mm in length and ~2 mm in width) at a flow rate of 0.2 mL h⁻¹. After 60 min, the solution from outlet and the overlays of toluene were collected. During the experiment, toluene was added to the system as needed to compensate for evaporation. The concentration of dye **2** in the aqueous phase was adjusted to 5 μg mL⁻¹ by dilution before taking UV–Vis spectra. More than 93.5% dye **1** was separated from the aqueous dye mixture into the toluene phase (Supplementary Fig. 10).

**All-liquid fluidic devices with wall-bound cationic small molecules**. An aqueous phase, consisting of cationic dyes (250 μg mL⁻¹) in a nanoclay dispersion (5 mg mL⁻¹), was pumped through the channel at a flow rate of 0.5 mL h⁻¹ for 12 min (Supplementary Fig. 11). After aging for 60 min, the channel was washed with water for 30 min at a flow rate of 0.5 mL h⁻¹. LSCM was conducted to determine the distribution of dyes in the channel (Fig. 3d and Supplementary Fig. 12).

**All-liquid fluidic devices with wall-bound cationic enzymes**. HRP[42] was first labeled with LRSC (Supplementary Figure 13a, b) prior to immobilization. Dye-conjugation was performed in an ice bath. Briefly, HRP (type VI, Sigma) was dissolved at 10 mg mL⁻¹ in borate buffer (50 mM, pH 9.0) before adding an aliquot of LRSC (10 mg mL⁻¹ in anhydrous N,N-dimethylformamide) at a molar ratio of dye-to-protein of 3:1. After 1 h, LRSC-labeled HRP was purified by spin dialysis to

remove the unreacted dye. LRSC-labeled HRP showed an emission maximum of 595 nm in 2-(N-morpholino)ethanesulfonic acid buffer (MES, 10 mM, pH 6.5) after excitation at 540 nm (Supplementary Fig. 13c).

To adsorb the enzyme onto the NPS wall, 100 μg mL$^{-1}$ of LRSC-labeled HRP in MES buffer (pH 6.5) was infused through the liquid channel at a flow rate of 0.5 mL h$^{-1}$ for 12 min. After aging for 60 min, the channel is washed by introducing pure MES buffer for 30 min at a flow rate of 0.5 mL h$^{-1}$ (Supplementary Fig. 13d). The distribution of HRP on the wall of the device was visualized using LSCM (Fig. 3e).

**All-liquid fluidic devices with wall-bound cationic nanocrystals**. CTAB-coated platinum nanocrystals (Pt NCs) were prepared according to a reported method with minor modifications[43]. Briefly, an aqueous solution of K$_2$PtCl$_4$ (1 mL, 10 mM) was mixed with an aqueous solution of CTAB (8.4 mL, 119 mM) in a 20-mL vial at room temperature. The mixture was then heated at 50 °C until the solution became clear, after which an ice-cold aqueous solution of NaBH$_4$ (0.6 mL, 500 mM) was added, and the reaction mixture stirred for 6 h at 50 °C. Excess CTAB was removed by centrifuging the solution twice at 14,000 × g for 30 min. The precipitate was collected and re-dispersed in 50 mL water. The size and size distribution of the NCs was observed using a Zeiss Gemini Ultra-55 analytical Field Emission Scanning Electron Microscope (Supplementary Fig. 14). A Malvern Zetasizer Nano instrument was used to measure the zeta potential.

Cationic CTAB-coated Pt NCs were immobilized onto the anionic NPS assembly and visualized directly at the liquid–liquid interface using an in-situ AFM technique (Fig. 3f). A minimal drop of silicon oil (viscosity $\eta = 60,000$ cSt) with 10% w/w of H$_2$N-PDMS-NH$_2$ was placed on a silicon substrate and 30 μL of nanoclay dispersion (10 mg mL$^{-1}$ in water) was added on top of the silicone oil to cover the oil completely. The sample was placed in a Cypher ES-sealed sample cell for 30 min, which allowed the nanoclay and polymer surfactants to assemble at the water–oil interface. Excess nanoclays were removed by exchanging the aqueous phase with water twice, after which the CTAB-coated Pt NC dispersion was added, allowing the NCs to adsorb the anionic NPS assembly through complementary electrostatics (Supplementary Fig. 15a). While imaging, the AFM tip was immersed in the NCs dispersion and brought directly into contact with the water–oil interface (Supplementary Fig. 15b, c).

**All-liquid microreactors-on-a-chip**. Using an all-liquid fluidic device configured with silicone oil (viscosity $\eta = 10$ cSt) as the oil phase, HRP in MES buffer (pH 6.5, 100 μg mL$^{-1}$) was pumped through the channel at a flow rate of 1 mL h$^{-1}$ for 10 min (Supplementary Fig. 16). After 15 min aging for enzyme immobilization, pure MES buffer was pumped through the channel at a flow rate of 1 mL h$^{-1}$ for 10 min to remove the free HRP. TMB (10 times diluted from commercial TMB from Sigma, pH 6.5) or 4-AAP/phenol peroxidase substrates (5 mM 4-AAP, 25 mM phenol, and 1 mM H$_2$O$_2$ in MES buffer, pH 6.5) were introduced. Two separate needles were connected to the inlet of the channel to avoid enzyme and substrate contamination. Conversion of substrates into colored products were observed along the length of the channel (Fig. 3g), which tracked color changes monitored exogenously in a cuvette (Supplementary Figs. 17, 18).

**Spatially programmable all-liquid microreactors for chemical logic**. Microchannels linking otherwise isolated regions on the patterned glass substrates were created using a variety of direct-write methods, obviating the need for additional surface patterns and allowing patterned regions to be differentially functionalized before connecting the system together as multicompartment system with compartmentalized functions. These functions, in turn, and by virtue of the reversibility of the printed connections, allowed the functions of the device to be programmed at will, spatially and temporally, rendering this all-liquid fluidic device as a programmable microreactor capable of carrying out chemical logic.

Using a Cellink Inkredible+ 3D Printer with a 14-gauge dispense tip as a print-head, we 3D-printed clay dispersions in silicone oil (viscosity $\eta = 60,000$ cSt) to generate 3D aqueous threads (Supplementary Fig. 20a). GCode was generated using a Python script. The print-head velocity was 300 mm min$^{-1}$, and flow rate was approximately 0.1 mL min$^{-1}$. If the gap between two separated channels was less than 5 mm, we printed the bridging arc manually (Supplementary Fig. 20b). By either method, bridging microchannels can be clipped, e.g., by using tweezers or scissors (Fig. 4c). The interconnectivity between patterned regions and the printed bridge was confirmed by pumping an aqueous dye solution (200 μg mL$^{-1}$ of resazurin dye in clay dispersion, pH 7) through patterned segments, which were connected by two liquid bridging arcs, at a flow rate of 0.15 mL h$^{-1}$ (Fig. 4d).

To realize the spatial and temporal control of catalyzed reactions in a programmed manner, we sequentially conducted three chromogenic reactions in the various channels under flow: oxidative coupling of 4-AAP with DAOS, oxidative coupling of 4-AAP with TOOS, and oxidative coupling of 4-AAP with phenol (Fig. 4e, Supplementary Fig. 21). The nanoclay dispersion (10 mg mL$^{-1}$) containing multiple reactants and catalyst were assigned to different 2D channels located on patterned substrate (inlet: 10 mM 4-AAP and 10 mM H$_2$O$_2$; CH1 branch: 100 mM DAOS and 80 μg mL$^{-1}$ HRP; CH2 branch: 25 mM TOOS and 20 μg mL$^{-1}$ HRP; CH3 branch: 25 mM phenol and 20 μg mL$^{-1}$ HRP; CH4 branch: 25 mM phenol). After fabricating bridges to connect the main road and certain branch

channel to turn-on the route, the 4-AAP/H$_2$O$_2$ solution was injected into the inlets at a flow rate of 0.15 mL h$^{-1}$ to trigger the chromogenic reactions to generate products with different colors under flow (Supplementary Fig. 20c). Bridges were cut off to prevent chemical contamination after finishing a desired reaction in one of the 2D channels. The conversion of substrates into various colored products in CHs 1–3 was consistent with expected color changes, e.g., when reactions were conducted in a cuvette (Fig. 4e). No color change was observed in CH4, where HRP was absent.

## Data availability
All data supporting the key findings of this study are available within the article and its Supplementary Information files or from the corresponding author upon reasonable request.

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

## Acknowledgements
This work was supported by the U.S. Department of Energy, Office of Science, Office of Basic Energy Sciences, Materials Sciences and Engineering Division under Contract No. DE-AC02-05CH11231 within the Adaptive Interfacial Assemblies Towards Structuring Liquids program (KCTR16). Portions of the work—including microscopy, device fabrication, and testing—were carried out at the Molecular Foundry, which is supported by the Office of Science, Office of Basic Energy Sciences, of the U.S. Department of Energy under Contract No. DE-AC02-05CH11231.

## Author contributions
B.A.H. and W.F. designed and B.A.H. directed the experiments. W.F. carried out pendant drop tensiometry, mass transfer experiments, LSCM, and all-liquid device fabrication and testing. Y.C. and P.D.A. carried out in-situ AFM experiments. J.F. and T.P.R. carried out mechanical characterization of NPS films and liquid-in-liquid 3D printing. W.F., J.F., T.P.R. and B.A.H. wrote the manuscript with contributions from all co-authors.

## Additional information

**Competing interests:** The authors declare no competing interests.

