## [Peer Review File · Nature Communications]

Reviewers' comments:

Reviewer #1 (Remarks to the Author):

I find this a fascinating topic and very interesting paper, however I am not very familiar with the use of NPSs for the purposes of creating quasi-rigid structures. I read several of the references as part of this review process and the advances beyond the already published work, by some of the same authors in many cases, is not clearly explained. Can the authors please provide a clear delineation between previous work and the current paper, especially in context of reference 6. Subject to satisfactory progress beyond existing work I recommend publication subject to the following aspects being addressed adequately,

1. In Figure 1(b) there are a range of concentrations given for interfacial tension and the variation with time, for the 0mg/ml case I assume the IFT is reduced due to the surfactant in the fluid, please provide the equilibrium value reached for this case.
2. Same figure, please explain why the IFT level is not the same for all conditions with nanoclay dispersions, I would have expected some equivalent CMC level to be reached at low concentrations and then the equilibrium interfacial tension becomes independent of concentration and function of time. Can you provide a measure of the effective "CMC" level for the nanoclay. If CMC is not relevant here then explain why the IFT levels vary with different concentration.
3. The existence of wrinkles on pendant drop experiments is known when using many different molecules, however my reading of this paper is that unless you see wrinkling there should not be an increased stiffness of the interface as the nanoclay will not have formed the required structure along the interface, however the channels of 1(e) do not appear to demonstrate any wrinkling effect, if this is the case then please explain what mechanism is driving the increased stiffness of the interface?
4. On your pendant drop experiments would it be possible to stop retraction of fluid and allow it to stabilize, to determine if this is irreversible or not (are the cases in supplementary figure 3 stopped or part of a movie). This wrinkling has been noted before but in those cases when the flow is stopped with a relatively large drop, the wrinkles disappear, hence in their cases a similar visual effect was at least reversible to some degree. A possible test of this stability would be to create the wrinkles then infuse into the drop again - if there is an elastic membrane on the drop, would this measure an effectively higher interfacial tension or some other variable using the pendant drop method? Please explain why does the formation of an elastic membrane reduce the interfacial tension in the first place; I would have thought if it is increasing stiffness then it would increase it but this may be my ignorance?
5. Line 116 – what is the maximum flow rate that you can achieve?
6. Line 129 – I think you need to provide some additional discussion about how the interface point with the needles seals, since it is now a metal boundary in a fluid. Does this require surfactants with certain properties, would plastic or other needles be the same?
7. Line 139 – "observed the rate at which dye 1 partitioned into the toluene phase", is this correct, you observed the reduction in the aqueous phase and inferred partitioning? Is there any mechanism that might allow you to quantify the dye in the toluene phase? Is it possible that the dye is going elsewhere, e.g. interface/nanoclays accumulation or even quenching over time?
8. Abstract and last paragraph around line 273 – I find the claims of applications here rather grand and not fully supported by the preliminary measurements in the text. They should be made more relevant or toned down a bit.
9. Overall, I find the relatively simple experiments are undertaken in a reasonable manner.
10. Figure 4 is amazing and looks beautiful.
11. Are there any caveats that are noteworthy for people considering using the technique? Would the authors suggest some possible disadvantages?

Reviewer #2 (Remarks to the Author):

This manuscript by Helms et al. describes a novel approach for creating millimeter cross section, all-liquid fluidic structures with use of nanoclay polymer surfactant assemblies. The key strength of this work is the novelty of making self-healing, dynamically modifiable fluidic features, with the potential to impact processes such as chemical reactions, catalysis and molecular analysis. This work could become suitable for publication in Nat. Comm., but revisions that may require additional experiments should be carried out. Detailed recommendations are enumerated below.

1. The fluidic structures created in this work, though potentially useful, are quite large in cross-sectional size, ranging from sub-millimeter to multiple millimeters across. Since the manuscript already describes structures having a range of sizes, it would be of interest to develop and describe scaling laws for making these channels. For example, could this approach be used to create nanofluidic features, having sub-micron cross sections? If not, what are the diameter constraints for these fluidic structures (10 microns? 100 microns?), since eventual applications will strongly depend on feature sizes that can be made. Additionally, since the vast majority of the fluidic structures described herein are in the millimeter diameter range, the word "microfluidic" should be replaced with "fluidic" in the title and throughout the manuscript.

2. The placement of Pt nanoparticles on the walls of these channels (Fig. 3c, f) seems a poor fit with the rest of the manuscript. And referring to these as "catalysts" when they have not been used as such detracts from the results with HRP that actually show catalysis. Fig 3c and f should be removed from the paper or more fully developed to demonstrate a catalysis application with Pt nanoparticles in these channels.

3. The authors compare the maximum volumetric flow in their system to that in ref. 21. Unless the fluidic features have the same cross sectional size, this information is essentially a reflection of the size of the channels in this work rather than their stability. The authors should compare the maximum linear flow rate to that in ref. 21 and other related all-liquid fluidic system work. It would also be helpful to compare the maximum linear flow rate in these systems with that of conventional closed microfluidic devices, such as PDMS.

4. The work described in Fig. 2 is a nice demonstration of liquid-liquid extraction by partitioning. The authors should avoid describing this result as a "separation", which infers something along the lines of liquid chromatography separation, which has not (yet) been shown in these systems.

5. The work builds on some much older literature from 15-20 years ago where the idea of all-liquid microfluidics was first explored, although it also references some recent work in this field (refs. 21 and 24). One related paper that should be cited is PNAS 2005 102, 9127-9132, which described a novel approach for creating much smaller, freestanding fluidic structures for DNA experimentation. However, with much of the all-liquid fluidics literature cited being less recent, the authors must clearly explain the significance of their advances in what some would consider to be a mature application space.

Reviewer #3 (Remarks to the Author):

The paper demonstrate a method to fabricate 3-dimensional microfluidic devices using liquid channels into another liquid phase. The channels are created using superhydrophobic-superhydrophilic micropatterned substrates and are stabilized by deposition of composite nanoclay-polymer surfactants at the liquid-liquid interface. The channels can be functionalized by deposition of small molecules, enzymes, and nanocrystals on the channel walls. Other application of the interface in the channels is as semi-permeable membrane to allow the exchange of components at the interface. Self-healing is an interesting property of the channels that allows them to change their connectivity and perform gating between different channels which opens the range of applicability of these devices.

The paper is excellent. My only comment is that usually the long wavelength shape perturbations known as Rayleigh and Plateau instability arise in cylindrical morphologies. Is it possible to ensure their long-term performance based on the channel and interfacial parameters? Since the interfacial tension is decreased when the NPS are adsorbed at the interface, how this affect the channel

stability since it is observed that the interfacial tension and contact angle are strongly affected by the presence of ions and other molecules in the fluids, in particular in cylindrical geometry, see Jimenez Angeles and Firoozabadi J. Physical J. Phys. Chem. C 120 (43), 24688-24696 (2016).

Reviewer 1: I find this a fascinating topic and very interesting paper, however I am not very familiar with the use of NPSs for the purposes of creating quasi-rigid structures. I read several of the references as part of this review process and the advances beyond the already published work, by some of the same authors in many cases, is not clearly explained. Can the authors please provide a clear delineation between previous work and the current paper, especially in context of reference 6. Subject to satisfactory progress beyond existing work I recommend publication subject to the following aspects being addressed adequately.

Author Response: We thank the reviewer for their favorable assessment of our work. In this manuscript, we report a new scheme to fabricate all-liquid microfluidic devices on micropatterned 2D substrates bearing superhydrophobic and superhydrophilic regions; we apply these patterned regions deliberately, to establish the system's wetting preference for the water and oil phases used in device construction and to control the chemistry in each isolated region. We further architect the device in 3D using liquid-in-liquid printing, where confined areas on the substrate can be connected by printed channels that are not pinned to the 2D substrate. This allows, for the first time, target regions of the 2D substrate to be differentiated in their functional attributes, either via the wall constituents or the channel constituents (or both). This is demonstrated in both Figures 3 and 4 in the main text. Thus, while the liquid-in-liquid printing on its own was reported by us in reference 6, here, it is a secondary tool to reach the expansive new property sets realized in all-liquid devices capable of performing useful work: chemical separations, chemical reactions, and colorimetric chemical screens toward spatially-programmable chemical logic.

This manuscript also brings to the forefront a new type of self-assembled wall, comprised of 2D nanoclays and polymers, that form quickly and in turn support the all-liquid architectures, allowing them to be printed broken and mended, and ultimately dictate the stability of the system under high-rate flow. The ability to flow at high rate was critical to functionalizing the walls of the device with catalysts, which had not been demonstrated previously, and further to use those catalysts to direct multi-step chemical transformations, which had also not been demonstrated. We also demonstrate chemo-selective transport small molecules across the wall, essentially providing a means to chemically couple the water and oil phases, e.g., where the chemical potential across the interface for various molecules can be used to drive a chemical separation. We hope that these considerations bring the reviewer greater appreciation of the advances described in this manuscript.

Reviewer 1: In Figure 1(b) there are a range of concentrations given for interfacial tension and the variation with time, for the 0mg/ml case I assume the IFT is reduced due to the surfactant in the fluid, please provide the equilibrium value reached for this case.

Author Response: The reviewer is correct. The assembly of nanoclay-polymer surfactant assemblies occurs very rapidly and is driven by the reduction in interfacial tension. In the absence of the nanoclay, the polymer on its own also acts as a surfactant to reduce the interfacial tension, however, not as significantly as when both polymer and nanoclay are present. In addition, with only the polymer surfactants, there is no stability of the printed constructs. To directly address the reviewer's request, we determined, using pendant drop tensiometry, the equilibrium IFT for a system configured with a water droplet suspended in toluene containing the polymer (10% w/w): $IFT = 18.9 \text{ mN m}^{-1}$ (**Supplementary Figure 3a**). In the associated figure caption, we direct the reader to the added benefit of the nanoclay in light of this control measurement taken in its absence.

New Figure Added in the SI:

Supplementary Figure 3 | Self-forming nanoclay–polymer surfactant assembly at the liquid–liquid interface. a, Temporal evolution of the interfacial tension (IFT) of an aqueous droplet, either loaded with the 2-D nanoclay (10 mg mL $^{-1}$, pH 7.0) (red) or not (purple), suspended in a solution of NH $_2$ -PDMS-NH $_2$ in toluene (10% w/w), differentiating the rates at which the system reaches steady-state and the extent to which nanoclay–polymer surfactants further decrease IFT when compared to the polymer surfactant layer on its own. Whereas an aqueous droplet suspended in toluene containing the polymer reached an equilibrium IFT of 18.9 mN m $^{-1}$ over 10 h; the nanoclay–polymer system reached steady-state with IFT of 12.8 mN m $^{-1}$ within 60 s. **b,** Snapshots over 12 h period of the nanoclay–polymer interfacial film’s irreversible buckling behavior, observed after retracting a cladded droplet that had reached steady-state with respect to IFT. Scale bar: 1 mm.

Reviewer 1: Same figure, please explain why the IFT level is not the same for all conditions with nanoclay dispersions, I would have expected some equivalent CMC level to be reached at low concentrations and then the equilibrium interfacial tension becomes independent of concentration and function of time. Can you provide a measure of the effective “CMC” level for the nanoclay. If CMC is not relevant here then explain why the IFT levels vary with different concentration.

Author Response: The steady-state interfacial tension (IFT) between water and oil with interfacially active species present will depend on their concentration, as these dictate the density of each at the interface. Reductions in IFT are commensurate with higher concentrations of interfacially active species. Concomitantly, in the approach to steady-state, higher concentrations of either (or both) lead to higher rates of nanoclay–polymer surfactant formation due to the increased rate of each component’s diffusion to the interface. Our nanoclay concentration-dependent IFT measurements were intended to quantify both the rapid kinetic trajectory of the nanoclay–polymer surfactant assembly as well as the differentiated steady-state for the concentration series in order to establish guidelines for which concentrations of nanoclay and polymer would be necessary for wall construction, both on the micropatterned substrates and during 3D liquid-in-liquid printing. In addition, the density of wall-forming species at the interface as well as the number of interactions between nanoclay particles and polymers are affected by the ionic strength of the aqueous phase for a given concentration of ionic wall-former, as informed from the reference (*Nano Letters* **2017**, *17*, 6453–6457). In the text, we now specify the role of concentration on these outcomes relevant to structuring liquids for microfluidic devices. We further direct the reader to

suitable references noting the effect in other classes of interfacial nanoparticle–polymer surfactant assemblies.

New Text Added in the text: *“The concentration of nanoclay influences the rate of diffusion to the interface, the ionic strength of aqueous phase and the areal density at the interface at steady-state (Supplementary Fig. 3a); the latter two ultimately dictate the IFT.”*

Reviewer 1: The existence of wrinkles on pendant drop experiments is known when using many different molecules, however my reading of this paper is that unless you see wrinkling there should not be an increased stiffness of the interface as the nanoclay will not have formed the required structure along the interface, however the channels of 1(e) do not appear to demonstrate any wrinkling effect, if this is the case then please explain what mechanism is driving the increased stiffness of the interface?

Author Response: The reviewer is correct in that the appearance of wrinkles is associated with the jamming of nanoclay–polymer surfactant assemblies at the interface. However, for wrinkling to occur, there must be a decrease in the interfacial area and the particles must be held firmly at the interface. We trigger the jamming of the nanoclay–polymer surfactant assemblies during the pendant drop experiments by retracting a small portion of the droplet’s internal volume to demonstrate unequivocally that an interfacial film has formed at steady-state and that, in its jammed state, the film is indefinitely stable (i.e., the wrinkles do not dissipate) as shown in Figure 1. In separate measurements, we intentionally jam nanoclay–polymer surfactant assemblies in a Langmuir trough to evaluate quantitatively, from their wrinkles, aspects of their mechanical properties, including their bending modulus (Supplementary Figure 5), which favorably distinguishes them from other classes of nanoparticle–polymer surfactant assemblies. Finally, we visualize using laser scanning confocal fluorescence microscopy the wrinkling behavior in all-liquid microfluidic devices (Figure 3 and Supplementary Figure 12), confirming in all cases that nanoclay–polymer surfactant assembly and jamming into an elastic film. Within the device, jamming is not triggered in the same manner as in the droplet or trough experiments, but rather due to the changes in internal channel pressure once the pumping is arrested and the system relaxes.

Reviewer 1: On your pendant drop experiments would it be possible to stop retraction of fluid and allow it to stabilize, to determine if this is irreversible or not (are the cases in supplementary figure 3 stopped or part of a movie). This wrinkling has been noted before but in those cases when the flow is stopped with a relatively large drop, the wrinkles disappear, hence in their cases a similar visual effect was at least reversible to some degree. A possible test of this stability would be to create the wrinkles then infuse into the drop again - if there is an elastic membrane on the drop, would this measure an effectively higher interfacial tension or some other variable using the pendant drop method? Please explain why does the formation of an elastic membrane reduce the interfacial tension in the first place; I would have thought if it is increasing stiffness then it would increase it but this may be my ignorance?

Author Response: In our system, the occurrence of jamming is irreversible due to the bifunctionality of the ligands. If the ligands were monofunctional and if there was no overlaying of the nanoclay sheets, the jamming would show a hysteresis, but it would be reversible. If the system is directed to create new interface, for example, by infusing droplets with additional liquid (not germane to the present study), or through the pumping of liquid through the channel at increasingly higher flow rates (which has been demonstrated extensively and noted in Figures 1, 3, 4, and Supplementary Figure 12), then the interfacial assembly would fracture or break, but the interfacial area that was created would be filled in with additional nanoclays and polymers. Once the area has been in-filled and a new steady-state reached, any jamming in the system will be irreversible as before. If the ligands are monofunctional or if exogenous means are used to disrupt the complementarity of interactions binding the polymer to the nanoparticle surface, then jamming can be reversed. We have explored those instances elsewhere (e.g., *Science*

Advances **2018**, 4, eaap8045; and *Advanced Materials* **2016**, 28, 6612). We now direct the reader to those studies to set the context for film formation.

In this work, it should be noted, we carry out all experiments where the electrostatic interactions between nanoclays and polymer surfactants promote assembly and jamming. As such, we observe no relaxation of the wrinkles, e.g., as shown for up to 12 h after triggering the jamming of nanoclay–polymer surfactants at the interface (Supplementary Figure 3b). In order for the wrinkles to relax, either the interfacial area must be increased, as in contracting and expanding the pendant drop, or the areal density of the assembly must be increased, requiring in-plane motion of the nanoclay sheets. The latter is prohibited by the bifunctionality of the ligand, since one ligand can be attached to two sheets. Finally, the fundamentals for why nanoparticle–polymer surfactant assembly lowers the interfacial tension are discussed in detail in our seminal paper on the concept (*Science*, **2013**, 342, 460).

In addition, with regard to the thermodynamic driving forces and mechanical properties of interfacial films, increasing clarity in recent years has been put forth in Hermans, *et al. Soft Matter* **2015**, 11, 8048 and Sagis, *et al. Rev. Mod. Phys.* **2011**, 83, 1367. When a solid film is formed at the interface, the surface tension decreases (otherwise the film would not assemble). However, this is only part of the story. The surface tension is derived from the thermodynamic equation of state of the system. What we must also consider are the rheological properties of the material formed at the interface (i.e., we now have a quasi-2D particle assembly with shear, Young's, and bending moduli). This makes the interface capable of supporting significant anisotropic surface stresses, to which the surface tension now makes only a small contribution). Hermans, *et al.* articulates this eloquently, where the (now tensor) surface stress can be broken into an isotropic component (the surface tension) and an anisotropic 2x2 tensor, which accounts for the material properties of the film.

New Figure Added in the SI:

b

Supplementary Figure 3 | Self-forming nanoclay–polymer surfactant assembly at the liquid–liquid interface. b, Snapshots over 12 h of the nanoclay–polymer interfacial film's irreversible buckling behavior, observed after retracting a cladded droplet that had reached steady-state with respect to IFT. Scale bar: 1 mm.

Reviewer 1: Line 116 – what is the maximum flow rate that you can achieve?

Author Response: The maximum flow rate of the all-liquid microfluidic devices depends both on the cross-sectional area of the channel as well as the channel architecture. In this revision, we relate to the reader that maximum flow rates will vary accordingly.

New Text Added in text: *“The maximum flow rate of the all-liquid microfluidic devices will depend both on the cross-sectional area of the channel as well as the channel architecture, and can be explored endlessly as such.”*

Reviewer 1: Line 129 – I think you need to provide some additional discussion about how the interface point with the needles seals, since it is now a metal boundary in a fluid. Does this require surfactants with certain properties, would plastic or other needles be the same?

Author Response: The reviewer brings up an excellent point, given the types of fluids used in the device and the need to make contact with the water phase for 3D printing channels and for pumping fluid in the device. We note that the clays are hydrophilic and the polymer ligands are surfactants, where the polar head groups can interact favorably with a hydrophilic needle surface, and the hydrophobic chain between the head groups can interact more favorably with a hydrophobic needle. We find that hydrophilic needles are preferred, but no explicit restriction as to how that might be achieved with different classes of materials, be they metals or plastics. We now offer an advisement to the reader.

New Text Added in SI: *“Stainless steel needles (22-gauge, 0.413 mm inner diameter) affixed to syringes, which were attached to syringe pumps, were inserted into the inlet and outlet arms of the circuits (Supplementary Figure 1c) to complete the device. Plastic or other needles are also applicable here, and needles with hydrophilic surfaces are preferred to realize the most effective seals.”*

Reviewer 1: Line 139 – “observed the rate at which dye 1 partitioned into the toluene phase”, is this correct, you observed the reduction in the aqueous phase and inferred partitioning? Is there any mechanism that might allow you to quantify the dye in the toluene phase? Is it possible that the dye is going elsewhere, e.g. interface/nanoclays accumulation or even quenching over time?

Author Response: We directly monitor the diffusion of ambipolar, non-ionic dye **1** from the aqueous phase to the toluene phase by time-dependent optical absorption (Figure 2b and Supplementary Figures S7a and 7b). There is no obvious reduction of the absorbance in the extinction spectrum for dye **1** in the aqueous phase in the presence of nanoclays, indicating the adsorption of dye **1** to the nanoclay can be ignored (Figure 2b). To provide evidence for this conclusion, we carried out an additional control experiment where we endeavored to monitor the adsorption of dye **1** to nanoclays in the aqueous phase in the absence of oil phase (Supplementary Figure 8). No obvious accumulation of the dye onto nanoclays was observed, neither was significant dye bleaching.

Contrasting this behavior, we show using the same experimental apparatus (Supplementary Figure 7a) that neither anionic nor cationic organic dyes crossover (Supplementary Figure 7c and 7d). We further provide evidence that cationic dyes adsorb to the anionic nanoclays, where time-dependent adsorption to the clay gives rise to distinct changes in the extinction spectrum of the cationic dye (Supplementary Figures 7d).

These cuvette experiments provide corroborating evidence for the in-line separation of dyes carried out in the all-liquid microfluidic device operating under flow (Figure 2c–e). Both the toluene phase and the aqueous phase at the end of the channel are analyzed by absorption spectroscopy to firmly establish the efficacy of dye partitioning into the toluene phase from within the channel. Our use of absorption spectroscopy is useful in that it provides these quantitative metrics without consideration to any fluorescence quenching of the dye. As a final demonstration of the dye partitioning, we use laser scanning confocal fluorescence microscopy to spatially and temporally monitor the partitioning, taking advantage

of the solvent-dependent optical properties of ambipolar dye **1**, which is fluorescent in the aqueous phase on excitation at 405 nm and less so in the toluene phase at the same wavelength due to shifts in the absorption maximum to higher wavelengths (Supplementary Figure 6b). To summarize, the interpretation of the LSCM data is aided by the solvatochromic behavior of the dye in polar vs non-polar solvents. We can selectively observe the aqueous phase by exciting the chromophore at longer wavelengths, where it selectively adsorbs. Finally, direct observation of the mass transfer of the neutral dye **1** cross the NPS wall is measured in real-time, not only from the LSCM data in Figure 2d and 2e, but also qualitatively in Supplementary Video 3.

New Text Added in SI: “Adsorption of non-ionic dye **1** to the nanoclays can be ignored (Supplementary Fig. 8).”

New Figure Added in the SI:

Supplementary Figure 8 | Tracking the non-ionic dye **1 in aqueous phase.** **a**, Stacked UV-vis spectra of non-ionic dye **1** at 10 min time-intervals in the absence of the oil phase. Neither the accumulation of dye **1** on the dispersed nanoclays nor dye quenching was observed over 6 h. **b**, Stacked UV-vis spectra of non-ionic dye **1** at 10 min time-intervals with oil phase, monitoring dye transfer from water to toluene across the NPS membrane film in the masked cuvette. **c**, Evolution in absorbance at $l = 438$ nm for an aqueous phase containing non-ionic dye **1** in systems configured with (**blue**) or without (**purple**) the oil phase, illustrating dye partitioning from water to toluene across the self-assembled nanoclay-polymer interfacial film.

Reviewer 1: Abstract and last paragraph around line 273 – I find the claims of applications here rather grand and not fully supported by the preliminary measurements in the text. They should be made more relevant or toned down a bit.

Author Response: In the abstract and conclusion, our aim is to stimulate the imagination for how our energetically-coupled all-liquid microfluidic devices may be applied in the future. Colorimetric assays, such as the tri-color assay demonstrated in Figure 4, are a clear indication that spatially programmable chemistries can report back local chemistry in any of the device's programmed regions. We also show that these colorimetric reactions can be triggered in any order by forming and then breaking channels in the device to establish specific flow paths on the substrate. In that solid-state fluidic devices can be configured with a variety of in-line detectors, it is quite reasonable that these can be coupled to the direct write and erase capabilities (i.e., 3-D liquid-in-liquid printing) of our devices and their reconfigurable network of channels to, in essence, process a chemical input detected in one region of the device and then autonomously reconfigure the flow paths to a new network configuration and thereby set in motion a different set of down-stream chemistries. While we have done this manually in the current proof-of-concept demonstration, the outward projection is that this can be automated via sensors, processors, and code to execute chemical logic from an integrated system. Re-Imaging apparatus for programmable and automated synthesis is an emerging intersection of chemistry, computation, and engineering (see, e.g., *Science* **2018**, eaav2211; *ACS Cent. Sci.* **2018**, *4*, 793; *Nature Commun.* **2018**, *9*, 2849; *ACS Cent. Sci.* **2018**, *4*, 144); these recent advances are now referenced to provide additional context for our statements. In addition, microfluidics are fairly standard devices for high-throughput screens for chemistry, materials, and biology, where chemical outputs serve as deep and dark data-sets for machine learning to create logic trees that accelerate the discovery and development of advanced pharmaceuticals, biologics, and materials. It is likely an all-liquid device such as ours provides an expanded set of capabilities in that regard. We hope this assuages the reviewer's initial concern.

Reviewer 1: Overall, I find the relatively simple experiments are undertaken in a reasonable manner. Figure 4 is amazing and looks beautiful.

Author Response: We thank the reviewer again for their favorable assessment of our work.

Reviewer 1: Are there any caveats that are noteworthy for people considering using the technique? Would the authors suggest some possible disadvantages?

Author Response: We agree with the reviewer that establishing guidelines for future successes and potential failures for those in the field is a worthwhile activity. To that end, we now note guidelines, which have come up in the course of our studies. These include mention of pH ranges for compatible aqueous buffers, flow rates, and range of channel dimensions that are patternable.

New Text Added in SI: *"In operating our all-liquid microfluidic devices, we find optimum pH ranges to be 5.0–9.0 for aqueous fluids pumped at optimum flow rates of 0.15–10 mL h⁻¹ for channel dimensions of 300–2000 μm."*

Reviewer 2: This manuscript by Helms et al. describes a novel approach for creating millimeter cross section, all-liquid fluidic structures with use of nanoclay polymer surfactant assemblies. The key strength of this work is the novelty of making self-healing, dynamically modifiable fluidic features, with the potential to impact processes such as chemical reactions, catalysis and molecular analysis. This work could become suitable for publication in *Nat. Comm.*, but revisions that may require additional experiments should be carried out. Detailed recommendations are enumerated below.

The fluidic structures created in this work, though potentially useful, are quite large in cross-sectional size, ranging from sub-millimeter to multiple millimeters across. Since the manuscript already describes

structures having a range of sizes, it would be of interest to develop and describe scaling laws for making these channels. For example, could this approach be used to create nanofluidic features, having sub-micron cross sections? If not, what are the diameter constraints for these fluidic structures (10 microns? 100 microns?), since eventual applications will strongly depend on feature sizes that can be made. Additionally, since the vast majority of the fluidic structures described herein are in the millimeter diameter range, the word "microfluidic" should be replaced with "fluidic" in the title and throughout the manuscript.

Author Response: We thank the reviewer for the suggestion to establish limits for the scaling of the microchannels. Prior art has firmly established the resolution of our micropatterns, which can be made as small as $\sim 10\ \mu\text{m}$ (see *Advanced Materials Interfaces* **2014**, *1*, 1400269). In this revision, we now make a note to the reader indicating these limits in the Supplementary Information, as was requested by Reviewer 1. In addition, the reviewer correctly points out that we have taken care in the manuscript to show that our devices can be made with channels varying in both architecture and dimension. One aspect that may or may not have been immediately appreciated is that, at the device dimensions investigated (Supplementary Figure 2), we are able to offer to the reader a full description of behaviors and definitively link the structure–property relationships of the nanoclay–polymer surfactant wall to the reconfigurable features of the all-liquid devices in both 2-D and 3-D. Nevertheless, we were successful in scaling of the devices to the micrometer range (Supplementary Figure 2) and the length-scale of the 3-D printed channels in Figure 4 are characteristically “micro”. For those reasons, we applied the microfluidic terminology throughout initially. In this revision, we revised the text to note it where physically demonstrated in a given fluidics experiment.

Reviewer 2: The placement of Pt nanoparticles on the walls of these channels (Fig. 3c, f) seems a poor fit with the rest of the manuscript. And referring to these as "catalysts" when they have not been used as such detracts from the results with HRP that actually show catalysis. Fig 3c and f should be removed from the paper or more fully developed to demonstrate a catalysis application with Pt nanoparticles in these channels.

Author Response: The reviewer may well be familiar with the catalytic properties of the various classes of catalysts successfully immobilized to the walls of our all-liquid devices. Germane to the present study, we were drawn to the peroxidase-like activity of Pt nanoparticles (see, for example, *Colloids and Surfaces A: Physicochemical and Engineering Aspects* **2011**, *373*, 6) and other nanoparticles (*Nature Nanotechnology* **2007**, *2*, 577). This sets up a competition, of sorts, between a natural peroxidase, HRP, and an artificial peroxidase, Pt nanoparticles, in carrying out the chemical transformations noted in Figures 3 and 4. To understand which had the highest activity, we conducted a range of experiments. Oxidative coupling of 4-AAP and phenol (5 mM 4-AAP, 25 mM phenol, 50 mM H_2O_2) catalyzed by Pt nanoparticles was conducted in a cuvette (**Supplementary Figure 16**). A product with light pink color was observed in the cuvette after adding Pt nanoparticles for 15 min, illustrating the peroxidase-like activity of Pt nanoparticles. By comparison, the color change of the 4-AAP/phenol substrate was much quicker and deeper when HRP was used as the catalyst (**Supplementary Figure 16c**). The superior activity of HRP was notable, and we carried that knowledge over to proof-of-concept demonstrations in the all-liquid fluidic device accordingly. In this revision, we make note of the superior activity of HRP enzymes relative to Pt catalysts in the devices.

New Text Added in SI: “Both HRP and CTAB-coated Pt NCs¹¹ have peroxidase activity. To understand which had the highest activity, we conducted a comparative study of the oxidative coupling of 4-aminoantipyrine (4-AAP) and phenol (5 mM 4-AAP, 25 mM phenol, 50 mM H_2O_2). The reaction catalyzed by Pt NCs was conducted in a cuvette (Supplementary Figure 16). The product of the reaction was colorimetrically confirmed (i.e., a light pink color) only after adding Pt nanoparticles (reaction time

= 15 min). By comparison, the color change was significantly deeper of using HRP for the same concentration of substrates and reagents (Supplementary Figure 16c). The superior activity of HRP was put to work in our proof-of-concept demonstrations that such reactions could be conducted in the micropatterned all-liquid fluidic device.”

New Figure Added in SI:

Supplementary Figure 16 | Peroxidase-like activity of CTAB-coated Pt nanoparticles. **a**, Schematic of the oxidative coupling of 4-AAP and phenol, catalyzed by natural and artificial peroxidases. **b**, Color change of a 4-AAP/phenol/H₂O₂ solution (5 mM 4-AAP, 25 mM phenol, 50 mM H₂O₂) over time after adding 40 μL of Pt NPs suspension ($\sim 100 \mu\text{g mL}^{-1}$). **c**, Photos of a 4-AAP/phenol/H₂O₂ solution (5 mM 4-AAP, 25 mM phenol, 50 mM H₂O₂) after adding 4 μL of HRP ($100 \mu\text{g mL}^{-1}$, pH 6.5, left photo) over 1 min or after adding 40 μL of Pt NPs suspension ($\sim 100 \mu\text{g mL}^{-1}$, right photo) over 60 min, showing the superior activity of HRP relative to CTAB-coated Pt nanoparticles in this reaction.

Reviewer 2: The authors compare the maximum volumetric flow in their system to that in ref. 21. Unless the fluidic features have the same cross sectional size, this information is essentially a reflection of the size of the channels in this work rather than their stability. The authors should compare the maximum linear flow rate to that in ref. 21 and other related all-liquid fluidic system work. It would also be helpful to compare the maximum linear flow rate in these systems with that of conventional closed microfluidic devices, such as PDMS.

Author Response: We agree with the reviewer that the comparison of the maximum linear flow rate to that in Ref 21 is more accurate here. Both the cross-sectional size and the architectures of the channels affect the maximum flow rate.

In general, the velocity profile of the fluid in the tube obeys Stokes law, i.e., the linear velocity in the tube is a parabolic curve. However, the data presented in Walsh *et al.* (*Nature Comm.* **2017**, 8, 816) (as we discuss later, this is the most appropriate system for a direct comparison) do not allow for a full quantitative comparison, given that the dependence of flow-rate through the channels varies as approximately $Q \sim R^4$, making measurements highly sensitive to errors in channel width. Channel length is, likewise, hard to obtain with accuracy from the Walsh paper (one expects $Q \sim R^{-1}$). We have modified the statement:

The original statement read:

"[...] an order of magnitude faster than the maximum reported in liquid microfluidics configurations that lack a functional interfacial membrane."

To now read:

"[...] significantly faster than the maximum reported in liquid microfluidics configurations that lack a functional interfacial membrane (as also shown in Fig. 1e and 1f)."

A number of observations justify this point:

1. Figs. 1e and f constitute the control experiment in which we explicitly demonstrate that our method allows for faster flow than that shown in, e.g., Walsh, *et al.* Flow rates of 10 mL h^{-1} can easily be achieved in our system containing a functional membrane, while in the absence of a membrane this flow rate results in growth of the footprint of the aqueous phase into the superhydrophobic region. Note that pinning at the heterogeneous contact line in Fig. 1 is likely stronger than in Walsh, *et al.* (though surface roughness, a critical factor, is hard to estimate in either case).
2. Walsh, *et al.* claim maximum flow rates of $300 \mu\text{L h}^{-1}$ in a channel of diameter approximately 1.5 mm (measured by assuming the bright region of the fluorescence image in Fig. 5b is the half-width of the channel). By contrast, in Fig. 1 we achieve 10 mL h^{-1} in a channel of diameter 2 mm or 0.5 mL h^{-1} in a channel of width 0.635 mm. Both measurements justify the "order-of-magnitude" statement in the original text assuming our measurements of channel width in Walsh, *et al.* are reasonable (assuming our estimates of channel width extracted via image analysis are accurate); though, we appreciate that measurements of channel length and diameter are challenging in this case, hence the above modification to the text.

Flow rate data in other works is scarce: Casavant, *et al.* (*PNAS* **2013**) do not report flow rates; Hsu, *et al.* (*Lab Chip* **2004**) use extremely small channels in a lithography-generated system and is not suitable for comparison (i.e., confinement is due to 3 solid walls) and lacks the functional advantages of our system; Juncker, *et al.* (*Anal. Chem.* **2002**) and Hong and Pan (*Microfluid Nanofluid* **2011**) use channels of radically different dimensions and geometries to our own; Gau, *et al.* (*Science* **1999**) use channels of radically different dimensions, do not discuss peak flow rates and, furthermore, point to another deficiency of surface microfluidics in the absence of the wall, i.e., the readiness with which differing channels can coalesce. By contrast, the nanoclay-polymer surfactant assemblies render systems highly resistant to coalescence (see, e.g., the wealth of literature on the superior coalescence-resistance of Pickering emulsions).

Flow rates in PDMS-based devices are far higher, $\sim \text{mL min}^{-1}$ (bulk elasticities will always beat surface elasticities). We can imagine a paper in which we engineer the cross-link density and assembly thickness to achieve higher flow-rates to do a full round-robin comparison of different microfluidic devices, though this would necessarily be a separate, extensive undertaking.

Reviewer 2: The work described in Fig. 2 is a nice demonstration of liquid-liquid extraction by partitioning. The authors should avoid describing this result as a "separation", which infers something along the lines of liquid chromatography separation, which has not (yet) been shown in these systems.

Author Response: In the revised manuscript, we now refer to the demonstration as an in-line purification via liquid–liquid partitioning.

Reviewer 2: The work builds on some much older literature from 15-20 years ago where the idea of all-liquid microfluidics was first explored, although it also references some recent work in this field (refs. 21 and 24). One related paper that should be cited is PNAS 2005 102, 9127-9132, which described a novel approach for creating much smaller, freestanding fluidic structures for DNA experimentation. However, with much of the all-liquid fluidics literature cited being less recent, the authors must clearly explain the significance of their advances in what some would consider to be a mature application space.

Author Response: We have added the citation to the list of references. We agree with the reviewer that the concept of liquid and surface microfluidics is not completely new, which is why we dedicate only part of Fig. 1 to liquid microfluidics qua liquid microfluidics (and place all extra material in the SI). Aside from the functional superiority of our device over other liquid demonstrated in our response to the previous comment, we are entirely unaware of any paper that combines this concept with a self-assembling, functional, elastic nanoparticle–polymer surfactant-based membrane. Moreover, we dedicate the overwhelming majority of our paper to demonstrate the applications-relevant practical benefits of this membrane:

Fig. 2 (chemical partitioning)

Fig. 3 (interfacially catalyzed chemical reactions with in-line extraction)

Fig. 4 In-situ reconfigurability and the compatibility with liquid 3D printing techniques, yielding 3D liquid microfluidic circuits with the capability for multiplexed chemical transformations.

Reviewer 3: The paper is excellent. My only comment is that usually the long wavelength shape perturbations known as Rayleigh and Plateau instability arise in cylindrical morphologies. Is it possible to ensure their long-term performance based on the channel and interfacial parameters? Since the interfacial tension is decreased when the NPS are adsorbed at the interface, how this affect the channel stability since it is observed that the interfacial tension and contact angle are strongly affected by the presence of ions and other molecules in the fluids, in particular in cylindrical geometry, see Jimenez Angeles and Firoozabadi *J. Phys. Chem. C* 120, 24688-24696 (2016).

Author Response: We thank the reviewer for their favorable view of our work. To address the reviewer's comment concerning Plateau–Rayleigh instabilities in liquids structured out of equilibrium into tubules, we have explored these in previous publications (*Nano Lett.* **2017**, *17*, 3119–3125; *Angew. Chem. Int. Ed.* **2017**, *129*, 12768–12772; and *Adv. Mater.* **2018**, *30*, 1707603). To summarize those findings, it is necessary to configure the system such that wall-forming is fast, and that the bending modulus of the jammed interfacial assembly formed is strong enough to counteract the compressive force in the system arising from the contraction of the interfacial area to minimize the energy. Nanoclays are a preferred embodiment of wall-formers, when used alongside the difunctional polymer, as noted in Figure 1, in Supplementary Figure 4, in Supplementary Figure 19, and in Supplementary Video 1.

We further agree with the reviewer that the mechanical strength of the assembly will be dictated by the density of wall-forming species at the interface, as well as the number of interactions between nanoclay particles and polymers. Both are affected by the ionic strength of the aqueous phase for a given concentration of ionic wall-former, as informed from the reference provided by the reviewer and quantified specifically for nanoparticle–polymer surfactants by us in a previous publication (*Nano Letters* **2017**, *17*, 6453–6457).

We should note that the influence of the underlying substrate stabilizes the system beyond these initial considerations of isolated tubules. However, for 3-D printed add-ons, the aforementioned physics apply when considering the stability over time. In our work, we have not observed the break-up or destruction of either the patterned 2-D channels or the 3-D printed inter-channel microtubes for devices in operation for up to 2 h. In this revision, we now note the observational windows for stability and expand the references to include these considerations of the underlying physics.

New Text Added in the text: *“The concentration of nanoclay influences the rate of diffusion to the interface, the ionic strength of aqueous phase and the areal density at the interface at steady-state (Supplementary Fig. 3a); the latter two ultimately dictate the IFT.”*

New Text Added in text: *“In that wall-forming is fast, the jammed assembly counteracts in a timely manner the compressive force in the system arising from Plateau–Raleigh instabilities:^{6,12,33} both patterned 2D channels and 3D printed inter-channel microtubes were stable for at least 2 h, i.e., while devices were in use.”*

Reviewers' comments:

Reviewer #1 (Remarks to the Author):

The authors have provided sufficient responses to most of this reviewer questions. I still do not understand the reasons around CMC equivalent to standard surfactant, and the authors additional point that the rate of diffusion increases is well known. The question is not about the rate but rather the end point regarding interfacial tension - that is when the surface is saturated should this apparent interfacial tension approach a constant? Of course this constant will be achieved at different time points depending on concentration.

Reviewer #2 (Remarks to the Author):

Overall the authors have been largely responsive to the review in their revisions, but there remain several things that should be addressed better before publication is warranted.

Reviewer 1: Line 116 – what is the maximum flow rate that you can achieve?

The authors' response was not useful: please estimate or report maximum linear flow rate for a standard linear architecture (which doesn't depend on all the factors mentioned in the response).

Reviewer 1: Abstract and last paragraph around line 273 – I find the claims of applications here rather grand and not fully supported by the preliminary measurements in the text. They should be made more relevant or toned down a bit.

This is the largest problem with this manuscript: it's a nice advance that is unfortunately somewhat negated by the authors' salesmanship. Please scale back or eliminate the unwarranted claims in these sections.

Reviewer 2: Additionally, since the vast majority of the fluidic structures described herein are in the millimeter diameter range, the word "microfluidic" should be replaced with "fluidic" in the title and throughout the manuscript.

The authors have not properly addressed this point. The scale bars in nearly all the figures are in the millimeter range. If the "microfluidic" terminology persists in the title and elsewhere, many readers of the paper will be disappointed at the disparity between is claimed in the title and abstract and what was actually done (see also the previous point).

Reviewer 2: The placement of Pt nanoparticles on the walls of these channels (Fig. 3c, f) seems a poor fit with the rest of the manuscript. And referring to these as "catalysts" when they have not been used as such detracts from the results with HRP that actually show catalysis. Fig 3c and f should be removed from the paper or more fully developed to demonstrate a catalysis application with Pt nanoparticles in these channels.

The authors haven't really addressed the point. The Pt nanoparticles were not demonstrated to have catalytic activity in the all-fluidic devices, like the HRP did. Justification is lacking for Fig. 3c, f (and the new Supplementary Fig 16 with cuvette data, rather than results from all-fluidic devices). Again, the above point about unnecessary salesmanship is relevant.

Reviewer #3 (Remarks to the Author):

The authors provided a detailed response to all the comments of the reviewers and adequately addressed the comments in the revised manuscript.

Reviewer 1: The authors have provided sufficient responses to most of this reviewer questions. I still do not understand the reasons around CMC equivalent to standard surfactant, and the authors additional point that the rate of diffusion increases is well known. The question is not about the rate but rather the end point regarding interfacial tension - that is when the surface is saturated should this apparent interfacial tension approach a constant? Of course this constant will be achieved at different time points depending on concentration.

Author Response: The density of polymers and nanoparticles at the interface at steady state is dependent on the concentration of each. Each nanoclay-polymer surfactant that forms leads to an iterative reduction in interfacial tension. Therefore the end-points are different; varying the concentration of nanoclay does not lead to the same endpoint faster, as suggested by Reviewer 1.

Reviewer 1: Abstract and last paragraph around line 273 – I find the claims of applications here rather grand and not fully supported by the preliminary measurements in the text. They should be made more relevant or toned down a bit. Please scale back or eliminate the unwarranted claims in these sections.

Author Response: We have revised the abstract and conclusion to address this concern.

Revised Abstract Text: “Multi-step chemical transformations can be conducted within the fluid channels under flow, as can selective mass transport across the liquid-liquid interface for in-line separations.”

Reviewer 2: What is the maximum flow rate that you can achieve?

Author Response: We have revised the manuscript to indicate that the maximum flow rate achieved was 10 mL h⁻¹.

Revised Text: “As a result, all-liquid fluidic systems may now be used at high flow rates (up to 10 mL h⁻¹) at markedly reduced surface tensions and, notably, in the presence of surface-active species.”

Reviewer 2: Additionally, since the vast majority of the fluidic structures described herein are in the millimeter diameter range, the word "microfluidic" should be replaced with "fluidic" in the title and throughout the manuscript.

Author Response: The revised manuscript now exclusively refers to the devices simply as fluidic devices.

Reviewer 2: The placement of Pt nanoparticles on the walls of these channels (Fig. 3c, f) seems a poor fit with the rest of the manuscript. And referring to these as "catalysts" when they have not been used as such detracts from the results with HRP that actually show catalysis. Fig 3c and f should be removed from the paper or more fully developed to demonstrate a catalysis application with Pt nanoparticles in these channels. The Pt nanoparticles were not demonstrated to have catalytic activity in the all-fluidic devices, like the HRP did. Justification is lacking for Fig. 3c, f (and the new Supplementary Fig 16 with cuvette data, rather than results from all-fluidic devices).

Author Response: We respectfully disagree with Reviewer 2's suggestion that we remove data from this manuscript concerning the Pt catalysts, which have been previously reported as artificial peroxidases. We show quite clearly that they are immobilized onto the wall and, ex-situ, that they are catalytically active;

they are simply not as active as HRP, which is why they are not explored in any depth in the fluidic devices. Supplementary Figure 16 accounts for the difference quite readily and convincingly, and we should provide to the reader the data that highlights those differences. To obfuscate these important details from the reader would be untoward.

Reviewer 3: The authors provided a detailed response to all the comments of the reviewers and adequately addressed the comments in the revised manuscript.

Author Response: We thank the Reviewer for their feedback.

REVIEWERS' COMMENTS:

Reviewer #1 (Remarks to the Author):

Reviewers comments addressed reasonably.

Reviewer #2 (Remarks to the Author):

The authors have responded appropriately to nearly all the referee comments. However, as stated in each previous round of review, the data on Pt nanoparticles embedded in these channels is peripheral to the manuscript, because the Pt nanoparticles add no demonstrated functionality to the fluidic channels or devices. And demonstrating ex-situ catalytic activity of particles is not relevant to their capabilities in fluidic channels, as the fluidic devices are the main point of the paper. Finally, the paper has sufficient novelty and technical merit for publication without including the peripheral data on embedding Pt nanoparticles in the channels.

I see two paths to making this manuscript acceptable for publication: (1) the authors can remove Figures 3c, f and Supplementary Figure 16 (and associated descriptive text regarding these figures), because none of these directly demonstrate that embedded Pt nanoparticles can be used as catalysts in these fluidic devices, or (2) the authors can demonstrate in-device catalysis with the embedded Pt nanoparticles.

Responses to Reviewers

Reviewer 1: Reviewers comments addressed reasonably.

Author Response: We thank Reviewer 1 for their service in the peer review process.

Reviewer 2: The authors have responded appropriately to nearly all the referee comments. However, as stated in each previous round of review, the data on Pt nanoparticles embedded in these channels is peripheral to the manuscript, because the Pt nanoparticles add no demonstrated functionality to the fluidic channels or devices. And demonstrating ex-situ catalytic activity of particles is not relevant to their capabilities in fluidic channels, as the fluidic devices are the main point of the paper. Finally, the paper has sufficient novelty and technical merit for publication without including the peripheral data on embedding Pt nanoparticles in the channels. I see two paths to making this manuscript acceptable for publication: (1) the authors can remove Figures 3c, f and Supplementary Figure 16 (and associated descriptive text regarding these figures), because none of these directly demonstrate that embedded Pt nanoparticles can be used as catalysts in these fluidic devices, or (2) the authors can demonstrate in-device catalysis with the embedded Pt nanoparticles.

Author Response: The Reviewer confirms that we demonstrate in our experimentation that HRP in solution outperforms Pt nanocrystal dispersions in peroxidase-type chemical transformations. These results are presented to and discussed with the reader, managing the reader's expectations for those classes of reactions for the different catalysts, were they used in a flow-driven microreactor. To directly address the reviewer's concern about the catalytic activity of Pt activity in peroxidase-type chemical transformations in a microreactor, we now include a strongly worded statement that the Pt nanocrystal catalysis in solution in no way correlates to the Pt catalysis in our device, and thus the activity may be different.

New Text Added: "Pt nanocrystal catalysis in solution in no way correlates to Pt catalysis in an all-liquid fluidic device, and thus the activity may be different."